# Prebiotics as adjunctive treatment ameliorates DSS-induced colitis and gut microbiota

Yan Kong,[1] Peitong Cao,[1] Jie Wu,[1] Xiaolin Ye[1]

**ABSTRACT** The disturbance of gut microbiota is closely related to the occurrence and development of inflammatory bowel disease (IBD), which can be improved by the supplementation of prebiotics. The therapeutic benefits of exclusive enteral nutrition (EEN) on patients with IBD are variable, and it has adverse effects on gut microbiota. Therefore, it is necessary to explore the efficacy of EEN combined with prebiotics on colitis and gut microbiota. In our study, mice were administered distilled water containing 4% dextran sodium sulfate (DSS) to induce colitis, then fed with normal water, EEN, EEN combined with isomaltooligosaccharide (IMO), EEN combined with fructo-oligosaccharide (FOS), and EEN combined with polydextrose (POL) for 1 week, respectively. The results revealed that EEN combined with prebiotic treatments showed more benefits in alleviating DSS-induced colitis than EEN treatment, among which FOS exhibited superior efficacy, including reducing disease activity index (DAI) (8.67 ± 0.52 vs 9.67 ± 0.82) and preventing colon shortening (7.23 ± 0.33 vs 6.43 ± 0.28 cm). Compared with the DSS group (7.00 [interquartile range, IQR: 5.75–8.25]), the histological score of IMO (2.50 [IQR: 2.00–4.00]), FOS (2.50 [IQR: 0.00–3.00]), and POL (2.00 [IQR: 1.50–7.25]) groups was obviously lower. In addition, prebiotics as adjunctive treatment increased more beneficial species than EEN treatment. Compared with the DSS group, POL adjuvant treatment enhanced the Simpson index, altered the overall structure of the gut microbiota, and IMO adjuvant treatment improved the F/B ratio (0.28 [IQR: 0.12–0.64] vs 1.06 [IQR: 0.87–2.82]). This study proved that prebiotics as adjuvant therapy improved colitis by regulating gut microbiota, providing a foundation for clinical application.

**IMPORTANCE** We demonstrated the superior efficacy of prebiotics as adjuvant therapy in improving clinical outcomes (body weight, colonic length, disease activity index, and histological score) and modulating the structure and composition of gut microbiota in dextran sodium sulfate-induced colitis in mice.

**KEYWORDS** prebiotics, gut microbiota, inflammatory bowel disease

Inflammatory bowel disease (IBD) is a chronic nonspecific inflammatory disease mainly involving the digestive tract. Crohn's disease (CD) and ulcerative colitis (UC) are two major subtypes, defined by both distinct and overlapping clinical and pathologic features (1). According to the global disease burden database, the number of patients with IBD increased from 3.32 million in 1990 to 4.9 million in 2019. The area with the highest prevalence rate is high-income North America, where the low-fiber and high-fat diet pattern is an important risk factor for IBD (2, 3). Although the pathogenesis of IBD is still unclear, it is suggested to be associated with the change of interaction between environment and intestinal microbiota, which leads to the abnormal immune response of genetically susceptible individuals (4).

The complex communication between intestinal microbiota and epithelial cells is critical in regulating intestinal homeostasis. More and more studies have shown that

**Peer Reviewer** Timmie Arthur Britton, Mayo Clinic Minnesota, Rochester, Minnesota, USA

Address correspondence to Jie Wu, wujiedoc@163.com, or Xiaolin Ye, 406302154@qq.com.

The authors declare no conflict of interest.

See the funding table on p. 11.

the disturbance of gut microbiota is closely related to the occurrence and development of IBD, which will adversely affect intestinal epithelial function and lead to the loss of immune tolerance. The composition and activity of gut microbiota in patients with IBD are distorted, characterized by decreased diversity, decreased proportion of Firmicutes, and increased proportion of Proteobacteria and Actinomycetes (5). Therefore, regulating intestinal microbiota is a potential treatment for IBD. Prebiotics are not digested by host enzymes and fermented by saccharolytic bacteria in the colon, which promotes the growth of beneficial microbial strains in the intestine (6). According to the number of monomers bound together, prebiotics can be divided into disaccharides, oligosaccharides, and polysaccharides (7). A large number of studies have shown that supplementation of prebiotics, including polydextrose (POL), fructo-oligosaccharide (FOS), and isomaltooligosaccharide (IMO), can alleviate colon inflammation by enhancing the integrity of intestinal epithelial barrier, inhibiting the expression of proinflammatory factors, regulating gut microbiota, and increasing the production of short-chain fatty acids (SCFAs). SCFAs not only fuel intestinal epithelial cells and boost intestinal barrier function but also act as ligands for G protein-coupled receptors, contributing to immune system activation and regulation (8).

Current treatments for IBD are not universally effective and have the potential for adverse events (9). Exclusive enteral nutrition (EEN) is not often used in UC patients, because it is less efficacious in these patients, which may be related to the lack of indigestible substrates in the main EEN formula, such as prebiotics. Although EEN is recommended as first-line therapy to induce remission in children with mild to moderate CD, up to 30% patients have no clinical response, and 50% patients have no endoscopic response to EEN (10, 11). In addition, most studies showed that EEN is correlated with reduced bacterial diversity and low levels of SCFAs, which may be due to EEN containing very few dietary components and no complex carbohydrates (12). A study on children with CD showed that the abundance of Bifidobacterium in feces decreased after EEN treatment (13). Therefore, EEN combined with prebiotics may be more effective in treating IBD. A study showed that enriching EEN with inulin-type fructans suppressed colitis development, increased relative the abundance of beneficial microbes, and led to an expansion of anti-inflammatory T-cell subsets (14). Therefore, the purpose of this study was to explore the effects of prebiotics as an adjuvant therapy for EEN on dextran sodium sulfate (DSS)-induced colitis and compare the efficacy of three prebiotics, POL, FOS, and IMO, providing a foundation for clinical application.

## MATERIALS AND METHODS

### Experimental animal grouping

Male specific-pathogen-free (SPF) C57BL/6 mice, aged 6–8 weeks, were selected for adaptive feeding in the SPF animal laboratory (room temperature about 25°C and humidity about 60%) for 1 week, during which they were free to eat and drink. Then randomly divided into six groups and numbered ($n = 6$ per group) (i). Control group received no additional intervention within 3 weeks. The remaining five groups were all administered distilled water containing 4% DSS for a duration of 7 days in the second week, then given different treatments in the third week: (ii) DSS group, given normal water to drink freely in the final week. (iii) EEN group, fed EEN in the final week. (iv) IMO group, fed EEN mixed with IMO, 0.3 mL each day in the final week. (v) FOS group, fed EEN mixed with FOS, 0.3 mL each day in the final week. (vi) POL group, fed EEN mixed with POL, 0.3 mL each day in the final week.

The enteral nutrition preparation in the EEN group was short peptide enteral nutrition propafenone (manufacturer: Nutricia Pharmaceutical Co., Ltd.). The preparation method was as follows: 50 mL cold water and 1 bag of propafenone were injected into a clean container, fully mixed and completely dissolved, then cold water was added to 500 mL to be stirred and mixed evenly. In other treatment groups, IMO (degree of polymerization is 3–7, manufacturer: Shanghai Yuanye Bio-Technology Co., Ltd.), FOS (degree of

polymerization is 3–7, manufacturer: Shanghai Yuanye Bio-Technology Co., Ltd.) or POL (degree of polymerization is 20, manufacturer: Shanghai Yuanye Bio-Technology Co., Ltd.) with a concentration of 20 mg/mL were added to the nutritional preparations respectively. The body weight and stool character of mice were recorded every day. On the 21st day, all mice were euthanized by cervical dislocation, performed under inhalation anesthesia with 1.5–3.0% isoflurane. The stool and colon tissue were collected for subsequent analysis.

## Disease activity index (DAI) and histopathological analysis

The DAI score was used to evaluate the severity of colitis. It was calculated by adding the scores of three parameters: weight loss, stool consistency, and presence of blood in the stool (15). The colon lengths were measured. Then they were rinsed with iced saline, fixed in 10% formalin, and embedded in paraffin. Tissue slices were prepared and stained with hematoxylin and eosin. The prepared samples were examined under a light microscope at 40× magnification. Histological scores were performed according to the severity of inflammatory cell infiltration, extent of damage, and degree of crypt damage (16).

## Gut microbiota analysis

### DNA extraction, PCR amplification, and sequencing

DNA from fecal samples was extracted by using a magnetic bead-based kit and quantified by Qubit dsDNA Assay Kit in Qubit 2.0 Fluorometer (Life Technologies, CA, USA). The genomic DNA was randomly sheared into short fragments. The obtained fragments were end repaired, A-tailed, and further ligated with Illumina adapter. The fragments with adapters were PCR amplified, size selected, and purified. The library was checked with Qubit and real-time PCR for quantification and bioanalyzer for size distribution detection. Quantified libraries will be pooled and sequenced on Illumina platforms, according to effective library concentration and data amount required.

### Bioinformatics analysis

Preprocessing the Raw Data obtained from the Illumina HiSeq sequencing platform using Readfq (V8, https://github.com/cjfields/readfq) was conducted to acquire the Clean Data for subsequent analysis. Considering the possibility of host pollution may exist in samples, Clean Data needs to be blasted to the host database which defaults to using Bowtie2.2.4 software (Bowtie2.2.4, http://bowtie-bio.sourceforge.net/bowtie2/index.shtml) to filter the reads that are of host origin. The MetaGeneMark (V2.10, http://topaz.gatech.edu/GeneMark/) software and CD-HIT software were used to predict the ORF. DIAMOND software (V0.9.9, https://github.com/bbuchfink/diamond/) was used to blast the Unigenes to the sequences of Bacteria, Fungi, Archaea, and Viruses which are all extracted from the NR database (Version: 2018-01-02, ) of NCBI with the parameter setting as blastp, with an $E$ value cut-off of 1e−5. Choose the result of which the $E$ value <the smallest $E$ value[10] to take the LCA algorithm which is applied to system classification of MEGAN software to make sure the species annotation information of sequences. Adopt DIAMOND software (V0.9.9) to blast Unigenes to functional database with the parameter setting of blastp, with an $E$ value cut-off of 1e−5. Differential microbiota was annotated by the Kyoto Encyclopedia of Genes and Genomes (KEGG) database to obtain the participating pathways. The alpha diversity index Shannon, Simpson, and Chao1 indices were calculated to measure the community richness and diversity. Principal coordinate analysis (PCoA) based on Bray-Curtis distance was used to clarify the differences among communities. Differential abundance analysis was conducted by linear discriminant analysis (LDA) effect size (LEfSe), and the false discovery rate was adjusted by the Benjamini-Hochberg method.

## Statistical analysis

Statistical analysis was performed using "R" software version 4.2.1. Continuous data were presented as mean ± standard deviation, or median (IQR) depending on the probability distribution. For comparisons between two groups, either Student's *t*-test or the Wilcoxon rank-sum test was employed. Multiple group comparisons were conducted using one-way ANOVA or Kruskal-Wallis *H*-test, then pairwise comparisons using the Mann-Whitney *U*-test were used to compare continuous variables. Statistical significance was set at $P < 0.05$.

## RESULTS

### Comparison of body weight, DAI, colonic length, and histological score in control, DSS, and treatment groups

One day before modeling (D7), there was no significant difference in body weight among the six groups of mice. At the end of modeling (D14), the body weight of DSS (16.93 ± 0.53 g), EEN (17.93 ± 1.17 g), IMO (17.71 ± 1.75 g), FOS (18.00 ± 0.96 g), and POL (17.36 ± 1.63 g) groups was significantly lower than the control group (22.50 ± 1.10 g) (all $P < 0.05$), and there was no significant difference among them (all $P < 0.05$). After 7 days of treatment (D21), compared with the DSS group (15.07 ± 0.61 g), the weight of EEN (17.09 ± 0.97 g), IMO (16.93 ± 1.77 g), and FOS (17.00 ± 0.71 g) groups increased significantly (all $P < 0.05$) (Fig. 1A). In terms of DAI, there was no significant difference among the five experimental groups at the end of modeling (D14). However, after 7 days of treatment (D21), the DAI of EEN (9.67 ± 0.82), IMO (9.17 ± 0.75), FOS (8.67 ± 0.52), and POL (9.83 ± 0.98) groups were significantly lower than the DSS group (12.00 ± 0.00). Especially, the DAI of the FOS group was significantly lower than the EEN group and the POL group (all $P < 0.05$) (Fig. 1B). On the last day of the experiment (D21), the colonic length and histological score of each group were compared. The results showed that the colonic length of EEN (6.43 ± 0.28 cm), IMO (6.75 ± 0.41 cm), FOS (7.23 ± 0.33 cm), and POL (6.70 ± 0.28 cm) groups were longer than the DSS group, while they were all shorter than the control group (7.838 ± 0.15 cm ) (all $P < 0.05$). In addition, the colonic length of the FOS group was longer than the EEN group ($P = 0.001$) (Fig. 1C and D). Compared with the DSS group (7.00 [interquartile range, IQR: 5.75–8.25]), the histological score of IMO (2.50 [IQR: 2.00–4.00]), FOS (2.50 [IQR: 0.00–3.00]), and POL (2.00 [IQR: 1.50–7.25]) groups was obviously lower (all $P < 0.05$), and there was no significant difference ($P = 0.099$) in the EEN group (3.50 [IQR: 3.00–5.00]) (Fig. 1E and F).

### Alterations of gut microbial diversity in control, DSS, and treatment groups

The alpha diversity of gut microbiota decreased due to DSS exposure. Compared with the control group, the Chao and ACE indices of the DSS group was obviously lower. Treatment with EEN did not improve the alpha diversity. Among the prebiotics adjunctive treatment groups, POL combined with EEN treatment reversed the decrease of Simpson index (Fig. 2A through D). DSS also had a significant impact on beta diversity, as shown in Fig. 2E. PCoA revealed a clear separation between the gut microbiota of the DSS and control groups, indicating an overall change in the gut microbiota structure. The gut microbiota structure of the EEN group was partially changed, as well as the IMO and FOS groups (Fig. 2E through G). Prominently, the POL group altered the overall structure of the gut microbiota affected by DSS exposure (Fig. 2H).

### Alterations of gut microbial composition in control, DSS, and treatment groups

We compared the differences of top 10 gut microbial compositions in the control, DSS, EEN, and EEN combined with prebiotics group at the levels of phylum, genus, and species, respectively. The main gut microbiota of the six groups was all *Bacteroidetes* and *Firmicutes* at the phylum level, but the ratio of their relative abundance was different (Fig.

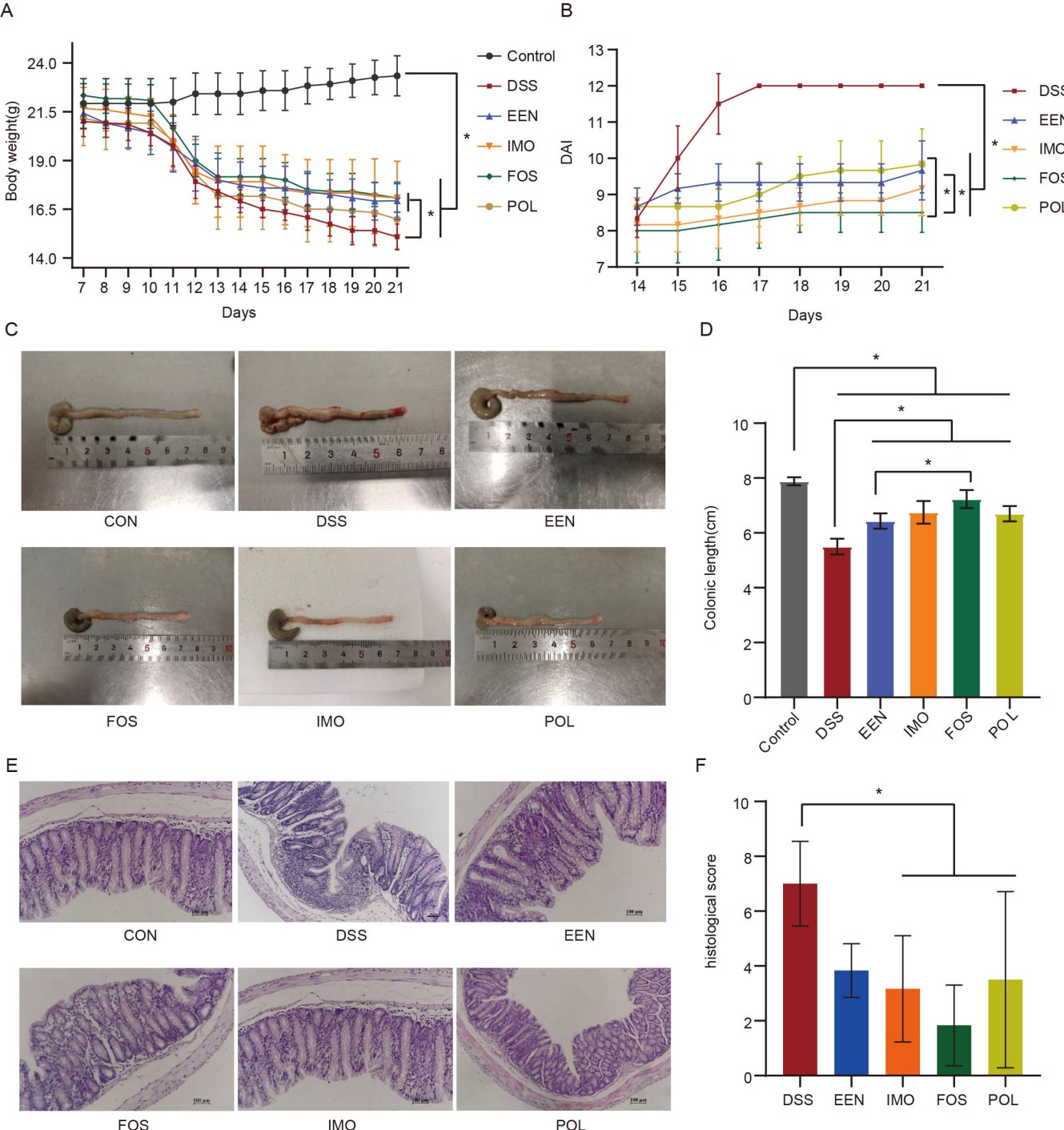

**FIG 1** Comparison of body weight (A), DAI (B), colonic length (C and D), and histological score (E and F) in control, DSS, and treatment groups.*P < 0.05. CON, control; DAI, disease activity index; DSS, dextran sodium sulfate; EEN, exclusive enteral nutrition; FOS, fructo-oligosaccharide; IMO, isomaltooligosaccharide; POL, polydextrose.

3A). Compared with the control group (1.25 [IQR: 0.68–1.88]), the *Firmicutes/Bacteroidetes* (F/B) ratio of the DSS group (0.28 [IQR: 0.12–0.64]) decreased significantly (*P* = 0.01). There was no significant difference in F/B ratio between the EEN group (0.77 [IQR: 0.16–1.61]) and the DSS group (*P* = 0.42). Among the prebiotics combined with EEN groups, IMO combined with EEN treatment (1.06 [IQR: 0.87–2.82]) reversed the decrease of F/B ratio induced by DSS exposure (*P* = 0.006) (Fig. 3B).

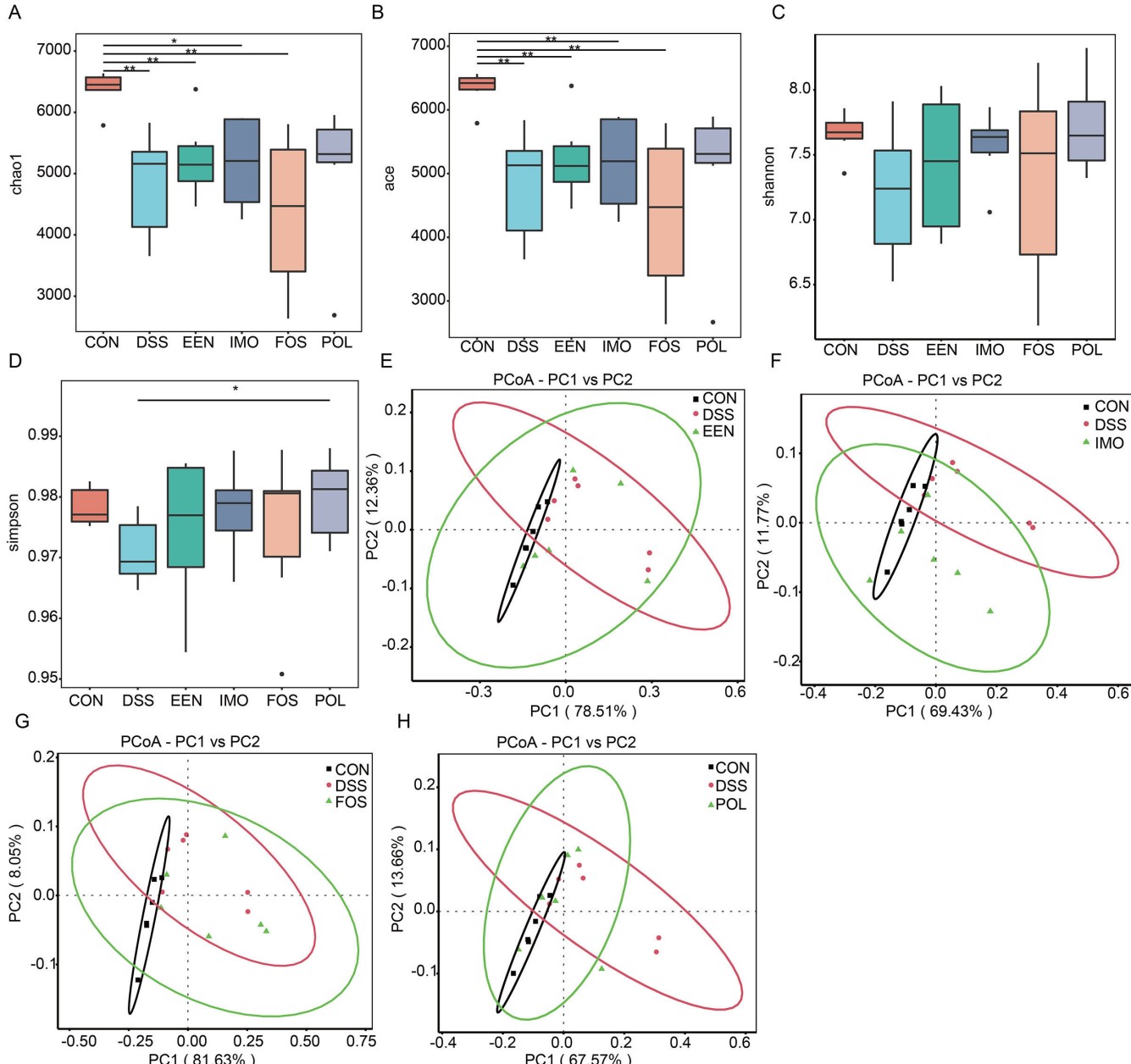

**FIG 2** Alterations of gut microbial diversity in control, DSS, and treatment groups. (A) Chao index. (B) ACE index. (C) Shannon index. (D) Simpson index. (E) PCoA analysis of control, DSS, and EEN groups. (E) PCoA analysis of control, DSS, and EEN groups. (F) PCoA analysis of control, DSS, and IMO groups. (G) PCoA analysis of control, DSS, and FOS groups. (H) PCoA analysis of control, DSS, and POL groups. *, $0.01 \leq P < 0.05$; **, $0.001 \leq P < 0.01$. CON, control; DAI, disease activity index; DSS, dextran sodium sulfate; EEN, exclusive enteral nutrition; FOS, fructo-oligosaccharide; IMO, isomaltooligosaccharide; POL, polydextrose.

At the genus level, the main gut microbiota in the control group was *Bacteroides* (9.73%), and its proportion of abundance in the DSS group increased significantly, accounting for 25.68%. In EEN, IMO, FOS, and POL treatment groups, the main gut microbiota was still *Bacteroides*, and the proportion of abundance of it was 21.82%, 15.48%, 31.52%, and 15.68%, respectively (Fig. 3C). At the species level, the main gut microbiota in the control group was *Lachnospiraceae Bacteroides* A4 (3.79%). In the DSS group, the main gut microbiota changed to *Bacteroidea* sp. CAG: 927 (3.75%), and the proportion of abundance of *Lachnospiraceae Bacteroides A4* decreased to 0.95%. In the treatment groups, the main gut microbiota in the EEN and POL groups remained

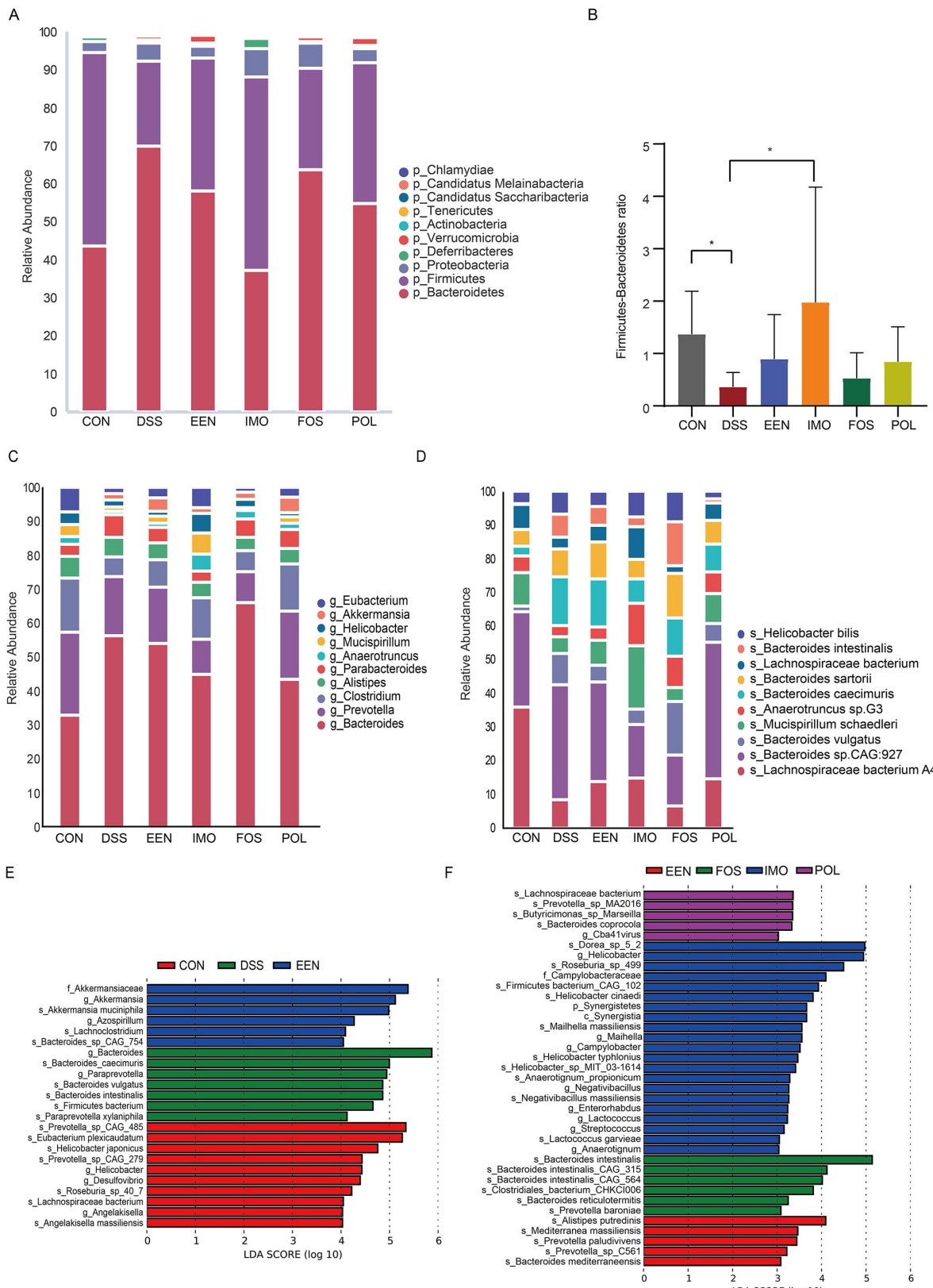

**FIG 3** Alterations of gut microbial composition in control, DSS, and treatment groups. The top 10 gut microbial compositions in different groups at the level of phylum (A), genus (C), and species (D). (B) *Firmicutes/Bacteroidetes* (F/B) ratio in different groups. (E and F) LEfSe analysis of the gut microbial community at the species level. *$P < 0.05$. CON, control; DSS, dextran sodium sulfate; EEN, exclusive enteral nutrition; FOS, fructo-oligosaccharide; IMO, isomaltooligosaccharide; POL, polydextrose.

unchanged, while the proportion of abundance of *Lachnospiraceae bacterium* A4 increased (1.48% and 1.11% respectively) (Fig. 3D).

LEfSe was used for discriminant analysis of multilevel species differences. The results showed that the abundance of several species belonging to *Bacteroides* increased caused by DSS exposure, including *Bacteroides caecimuris*, *Bacteroides vulgatus*, *Bacteroides intestinalis*, *Bacteroides dorei*, *Bacteroides ovatus,* and *Bacteroides xylanisolvens*. At the same time, the abundance of SCFA-producing related bacteria decreased, including *Roseburia* sp. 407 and *Lachnospiraceae bacterium*. After EEN treatment, the abundance of *Akkermansia muciniphila* and *Lachnoclostridium* increased significantly (Fig. 3E). We further compared the differences in intestinal species treated with EEN combined with or without prebiotics. Compared with the EEN group, the abundance of *Dorea* sp. 52, *Roseburia* sp. 49, *Firmicutes bacterium* CAG 102, all belonging to phylum *Firmicutes*, and *Lactococcus garvieae* increased significantly in the IMO group. The abundance of *Clostridiales bacterium* CHKCI006 and *Prevotella baroniae* in the FOS group was higher than the EEN group. Compared with the EEN group, the abundance of *Lachnospiraceae bacterium*, *Prevotella* sp. MA2016, and *Butyricimonas* sp. Marseilla in the POL group increased obviously (Fig. 3F).

## Difference in metabolic pathways in control, DSS, and treatment groups

The shared gene numbers of six groups were 482,566 accounted for 73.7% of the total 654,492 genes (Fig. 4A), indicating that there were some differences in different treatment groups. KEGG pathway enrichment analysis was further analyzed to assess the related functions and enrichment pathways of these different genes, and the relative abundance of pathways among groups was compared. This study found that the primary metabolic pathways in the IMO group were mainly concentrated in environmental information processing and cellular processes, and in the POL group were mainly concentrated in Genetic Information Processing (Fig. 4B). The differences in secondary metabolic pathways were as follows: genetic information transcription in the EEN group, signal transduction and membrane transport in the IMO group, cell process, glycan and lipid metabolism in the FOS group, xenobiotics and nucleotide metabolism in the POL group (Fig. 4C).

## DISCUSSION

Prebiotics play a beneficial role in IBD through various mechanisms, including providing a better breeding space for beneficial microbiota, inhibiting the growth of pathogens, and increasing the production of SCFAs (17). However, at present, there are few studies on the efficacy of prebiotics as adjuvant therapy for EEN by using animal models of colitis or patients with IBD, and the comparison of different prebiotics is lacking. By comparing the symptom, histological score, diversity, and composition of gut microbiota in DSS-induced colitis model treated with EEN and EEN combined with three different prebiotics, this study proved the benefits of prebiotics as adjuvant therapy and revealed its regulatory effect on gut microbiota.

In our study, EEN combined with prebiotic treatments showed more benefits in improving DSS-induced colitis than EEN treatment, including reducing DAI and histopathological score, preventing weight loss, and colon shortening that were elevated due to DSS exposure. In particular, the histopathological score was only decreased significantly after EEN combined with prebiotic treatments. We compared the efficacy of three different prebiotics, FOS, POL, and IMO, as adjuvant treatment, among which FOS exhibited superior efficacy. Compared with the EEN group, the DAI score was lower and the colon length was longer in the FOS group. A previous study also showed that FOS supplementation can significantly reduce the weight loss and histological damage of DSS-induced colitis in mice. In addition, FOS supplementation was observed to counteract the loss of tight junction proteins, reduce the expression of proinflammatory cytokines, and restore altered gut microbiota composition in mice with colitis (18).

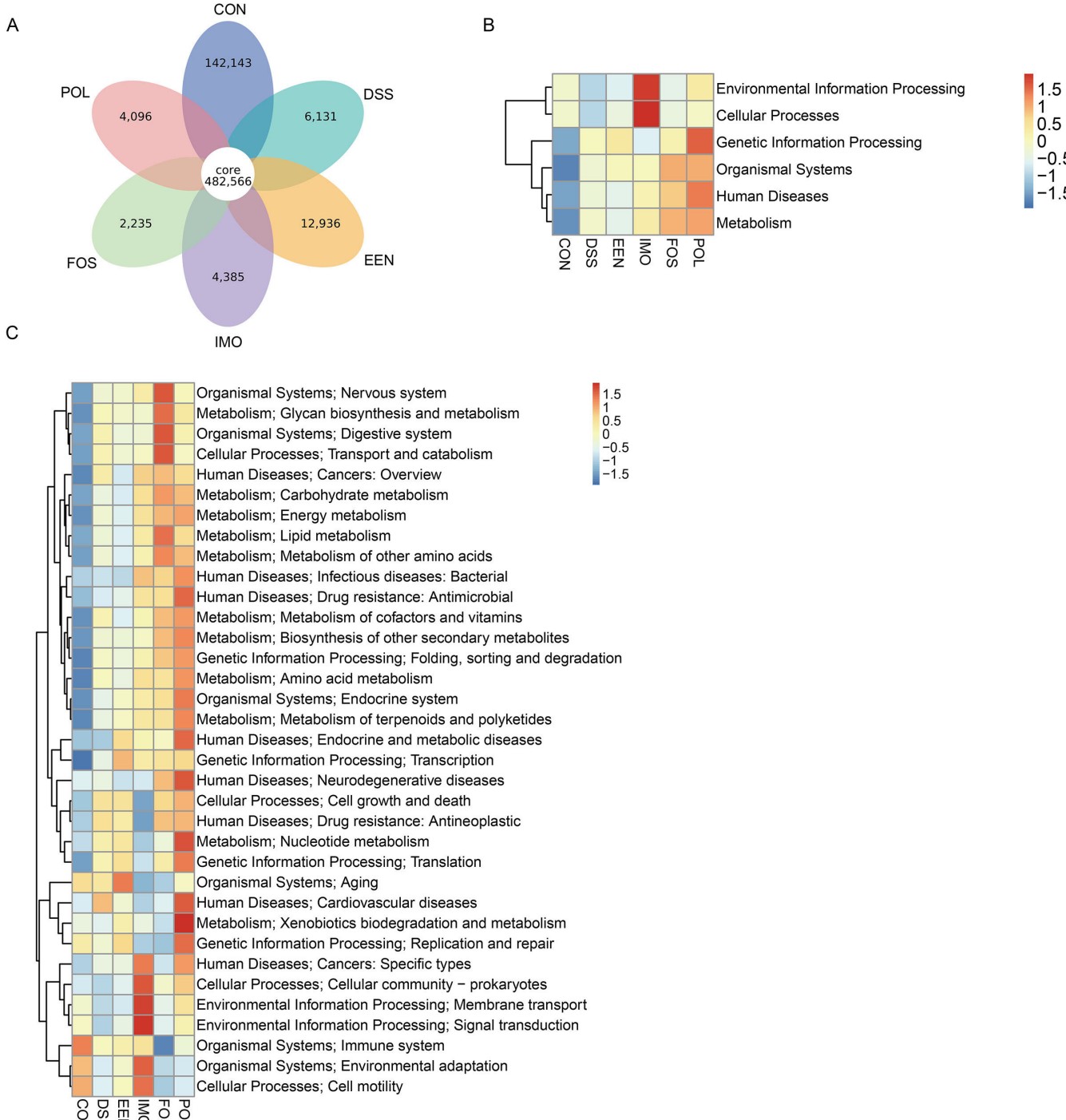

**FIG 4** Differences in genes and metabolic pathways of gut microbiota in control, DSS, and treatment groups. (A) Venn diagram of microbial genes. The differences between primary metabolic pathways (B) and secondary metabolic pathways (C). CON, control; DSS, dextran sodium sulfate; EEN, exclusive enteral nutrition; FOS, fructo-oligosaccharide; IMO, isomaltooligosaccharide; POL, polydextrose.

DSS exposure changed the diversity and the composition of gut microbiota. The F/B ratio of the DSS group was lower than the control group. In the gastrointestinal tract, Firmicutes and Bacteroidetes account for about 90% of the total microbiota (9). Many studies have also shown that patients with IBD are usually accompanied by gut microbiota disturbance, among which the phylum *Firmicutes* is reduced while *Bacteroidetes* and *Actinomyces* are increased (19). The F/B ratio has been associated with

maintaining homeostasis, and the increase of this ratio can lead to bowel inflammation. *Firmicutes* bacteria play an important role in the nutrition and metabolism through SCFA synthesis. The components of Bacteroidetes bacteria, lipopolysaccharides and flagellin, interact with cell receptors and enhance immune reactions through cytokine synthesis (20). A study on mucosal biopsy and RNA extraction of patients with IBD showed that compared with controls, the F/B ratio was significantly decreased in both UC and CD patients (21). In addition, the abundance of several species belonging to *Bacteroides* increased in the DSS group, which was associated with colitis (22), and the abundance of some beneficial bacteria decreased, including *Prevotella* sp. CAG: 485, *Prevotella* sp. CAG: 279, *Roseburia* sp. 407, and *Lachnospiraceae bacterium*. *Prevotella* plays a protective role against gut inflammation (23). Previous studies demonstrated that *Roseburia* improves the host's intestinal environment and activates the mucosal immune system (24). *Lachnospiraceae* members selectively colonize the mucus layer and produce the SCFAs (25).

We analyzed the gut microbiota of DSS-induced colitis in mice, thereby further exploring the mechanism of EEN and EEN combined with prebiotic. The diversity of gut microbiota did not enhance, while the abundance of *Akkermansia muciniphila* and *Lachnoclostridium* increased significantly after EEN treatment. A number of studies have shown that *Akkermansia* can improve mucus thickness and thus gut barrier integrity (26). *Lachnoclostridium* could synthesize butyrate via the 4-aminobutyrate/succinate pathway (27). The prebiotic adjuvant treatments ameliorated the microbiota in more aspects. POL adjuvant treatment reversed the reduction in Simpson index, altered the overall structure of the gut microbiota, and IMO adjuvant treatment improved the F/B ratio. The abundance of *Lachnospiraceae bacterium* A4 increased after EEN treatment, IMO, and POL adjuvant treatment.

Compared with EEN treatment, prebiotics adjuvant treatment increased more beneficial species. The abundance of *Dorea* sp. 52, *Roseburia* sp. 49, *Firmicutes bacterium* CAG: 102*,* and *Lactococcus garvieae* increased significantly in IMO adjuvant treatment. The *Dorea* belongs to the *Lachnospiraceae* family, and its deficiency was associated with the risk of IBD (28). *Lactococcus* exerted a protective effect on DSS-induced colitis in mice (29). FOS adjuvant treatment increased the *Clostridiales bacterium CHKCl006*, belonging to the class *Clostridia*, order *Clostridiales*, and produce SCFAs (30). The beneficial species with increased abundance in POL adjuvant treatment were *Lachnospiraceae Bacillus*, *Prevotella* sp. Ma2016, and *Butyricimonas* sp. Marcella. Quantitative analysis revealed a decrease of butyrate producers *Butyricimonas* in patients with IBD (31). A study showed that *Butyricimonas* had a negative correlation with pro-inflammatory cytokines, indicating that it may be effective in the progression of colitis (32).

In addition, KEGG pathway enrichment analysis showed that the relatively prominent pathways in the POL group were xenobiotics and nucleotide metabolism, while cellular processes, glycan, and lipid metabolism were relatively prominent pathways in the FOS group. Previous studies have also shown that the consumption of prebiotics not only affects the composition of the intestinal microbiome, but also changes its metabolic activity (33).

Nevertheless, this study had some limitations. Although animal models can replicate human diseases, the immune systems of animals and humans are not exactly the same, and their responses to prebiotics may be different. Therefore, the therapeutic efficacy of prebiotics combined with EEN in human IBD needs to be explored in future clinical trials. In addition, the types, dosage, and safety of prebiotics need to be further evaluated.

In summary, this study indicated that prebiotics as adjuvant therapy showed superior efficacy in ameliorating DSS-induced colitis, including preventing colon shortening, reducing DAI and histopathological scores, as well as improving the diversity and composition of gut microbiota.

## ACKNOWLEDGMENTS

This work was supported by the Natural Science Foundation of Beijing, China (7244340), the Natural Science Foundation of Beijing, China (J230009), the High-Level Public Health Technical Personnel Project (Academic leader-02-04), the Sanming Project of Medicine in Shenzhen (SZSM202311023), and the Beijing Hospitals Authority's Ascent Plan (DFL20221003).

Y.K., J.W., and X.Y. designed the study and wrote the manuscript. Y.K. and P.C. performed the experiments. Y.K. and X.Y. performed the statistical analysist. J.W. and X.Y. revised the manuscript. All authors contributed to the article and approved the submitted version. Y.K.: Conceptualization, Investigation, Methodology, Writing—original draft, Writing—review and editing. P.C.: Investigation, Methodology. J.W.: Conceptualization, Methodology, Writing—review and editing, Project administration. X.Y.: Conceptualization, Methodology, Writing—review and editing, Project administration.

## AUTHOR AFFILIATION

[1]Department of Gastroenterology, Beijing Children's Hospital, Capital Medical University, National Center for Children's Health, Beijing, China

## AUTHOR ORCIDs

Jie Wu  http://orcid.org/0000-0003-4028-6173
Xiaolin Ye  http://orcid.org/0000-0003-0086-0601

## FUNDING

| Funder | Grant(s) | Author(s) |
| --- | --- | --- |
| Natural Science Foundation of Beijing, China | 7244340 | Xiaolin Ye |
| Beijing Natural Science Foundation | J230009 | Jie Wu |
| High-Level Public Health Technical Personnel Project | Academic leader-02-04 | Jie Wu |
| Sanming Project of Medicine in Shenzen | SZSM202311023 | Jie Wu |
| Beijing Hospitals Authority's Ascent Plan | DFL20221003 | Jie Wu |

## DATA AVAILABILITY

The raw data presented in this study can be found in the NCBI repository with accession number PRJNA1242976.

## ETHICS APPROVAL

All the experimental protocols in this study were approved by the Institutional Animal Care and Use Committee of Capital Medical University, ethical approval number AEEI-2024-244.

## ADDITIONAL FILES

The following material is available online.

### Open Peer Review

**PEER REVIEW HISTORY (review-history.pdf).** An accounting of the reviewer comments and feedback.

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
