## [Reviewer comments · Microbiology Spectrum]

Microbiology Spectrum

Prebiotics as adjunctive treatment ameliorates DSS-induced colitis and gut microbiota

Yan Kong, Peitong Cao, Jie Wu, and Xiaolin Ye

Corresponding Author(s): Xiaolin Ye, Beijing Children's Hospital Capital Medical University

Review Timeline:

Submission Date:	May 14, 2025
Editorial Decision:	September 14, 2025
Revision Received:	October 29, 2025
Accepted:	November 20, 2025

Editor: Xue Zhang

Reviewer(s): Disclosure of reviewer identity is with reference to reviewer comments included in decision letter(s). The following individuals involved in review of your submission have agreed to reveal their identity: Timmie Arthur Britton (Reviewer #2)

Transaction Report:

DOI: <https://doi.org/10.1128/spectrum.01502-25>

Re: Spectrum01502-25 (Prebiotics as adjunctive treatment ameliorates DSS-induced colitis and gut microbiota)

Dear Dr. Xiaolin Ye:

Thank you for the privilege of reviewing your work. Below you will find my comments, instructions from the Spectrum editorial office, and the reviewer comments.

There are the comments from the first reviewer. We are still waiting for those from the other reviewer. Please find the comments at the end of this email.

Revision Guidelines

Sincerely,
Xue Zhang
Editor
Microbiology Spectrum

Reviewer #2 (Comments for the Author):

N/A

Reviewer #4 (Comments for the Author):

General comments:

This study aims to explore the effects of prebiotics as an adjuvant treatment to exclusive enteral nutrition (EEN) on DSS (sodium dextran sulfate) induced colitis and gut microbiota. The objective is to compare the efficacy of three prebiotics (FOS : Fructooligosaccharide; POL : Polydextrose; IMO : Isomaltooligosaccharides) to improve colitis symptoms, modulate the composition and diversity of the gut microbiota. The study was performed on C57BL/6 mice to evaluate the effects of prebiotics. The mice were divided into six groups, including a control group and five DSS-treated groups. The treated groups received normal water, EEN, or EEN combined with prebiotics (IMO, FOS, POL). The study lasted a total of 21 days, including one week of adaptation, one week of treatment with DSS to induce colitis, and one week of treatment with the different interventions. Body weights, stool consistency, and colon length were measured. Histopathological analyses were performed to assess inflammation. Microbiota analyses revealed that adjuvant treatments with prebiotics (FOS, POL, IMO) improved the diversity, composition and structure of the gut microbiota, in particular by increasing beneficial species and modulating diversity indices such as the Firmicutes/Bacteroidetes ratio.

Major concerns :

The abstract could be made more quantitative by including specific numerical data to support the results. For example, it could mention percentages of reduction in the Disease Activity Index (DAI), differences in colon length, or variations in microbial diversity indices (such as the Simpson index or the F/B ratio). This would better illustrate the magnitude of the effects observed and make the conclusions more convincing for the reader.

Materials and methods : Detailed information on the origin, type, and source of the prebiotics used in the study is missing. Although the introduction mentions the names of the prebiotics (polydextrose, fructooligosaccharides and isomaltooligosaccharides) and their beneficial effects, it does not provide details on the exact type : Although the names of the prebiotics are given, there is no detailed description of their specific characteristics, such as their chemical structure or degree of polymerization, which could influence their effectiveness and the source or supplier: The study does not mention whether these prebiotics were purchased from a specific supplier or prepared in a laboratory, which is important information for the reproducibility of the results.

Specific comments:

Lines 24-25 : The destruction of gut microbiota is closely related to the occurrence and development of inflammatory bowel disease (IBD), which can be improved by the supplementation of prebiotics.

Comments : The term "destruction" may not be the most accurate in this context. It would be more appropriate to replace it with "disturbance" or "dysbiosis," as these terms better describe the imbalance or alteration in the gut microbiota composition associated with inflammatory bowel disease (IBD). The revised sentence would read:

"The disturbance (dysbiosis) of gut microbiota is closely related to the occurrence and development of inflammatory bowel disease (IBD), which can be improved by the supplementation of prebiotics."

Reference

Dysbiosis (disruption of the normal gut microbiota) is a hallmark of IBD, characterized by reduced bacterial diversity, decreased beneficial bacteria, and increased potentially harmful bacteria.

Hu, Y.; Chen, Z.; Xu, C.; Kan, S.; Chen, D. Disturbances of the Gut Microbiota and Microbiota-Derived Metabolites in Inflammatory Bowel Disease. *Nutrients* 2022, 14, 5140. <https://doi.org/10.3390/nu14235140>

Line 38 : This study proved that probiotics as adjuvant therapy improved ... Please replace probiotics by prebiotics

Line 41 : Importance

Line 42 -44: Both sentences have some redundancy, as they both describe the benefits of prebiotics as an adjuvant treatment, albeit from slightly different angles. The first sentence focuses on clinical effects (body weight, colon length, ICD, histological score), while the second sentence focuses on microbiological effects (structure and composition of the gut microbiota).

Suggestion :

"We demonstrated the superior efficacy of prebiotics as adjuvant therapy in improving clinical outcomes (body weight, colonic length, DAI, and histological score) and modulating the structure and composition of gut microbiota in DSS-induced colitis in mice.

Minor comments :

The names of microorganisms are not always presented in italics, as is required by scientific conventions.

According to scientific standards, the names of genera and species of microorganisms should always be written in italics to respect taxonomic conventions.

Line 285 : *Prevotella*, line 286 : *Roseburia* Line 306 : *Lachnospiraceae* *Bacillus*, *Prevotella* SP Ma2016, and *Butyricimonas* SP Marcella; line 308 : *Butyricimonas*

References:

The quality of the references is as important as the quality of the manuscript itself and indicates the special care taken by the authors to submit a manuscript free of errors.

Reference section filled with errors (failure to italicize the scientific names of bacteria; article titles should be written in sentence style, without the capital letters throughout).

Suitable Quality? Yes

Sufficient General Interest? Yes

Clearly Written? Yes

Procedures Described? Yes

Supplemental Material Warranted? N/A

Summary: Here, Kong and colleagues set out to demonstrate exclusive enteral nutrition (EEN) reduces DSS-induced colitis in mice following supplementation with prebiotic carbohydrates – isomaltooligosaccharides (IMO), fructooligosaccharides (FOS), and polydextrose (POL). Authors note EEN treatment is less efficacious in ulcerative colitis (UC) patients, possibly due to the lack of fermentable substrates. While the authors claim these prebiotic adjuvants improve body weight, disease activity index (DAI), histopathology score, and colonic length compared to EEN treatment alone, the effects were minimal or often non-significant, despite providing clear benefits compared to DSS-treatment alone. In addition, there were many instances where bacterial names were not italicized, misspelled, or the wrong term used. The study significance could be improved with more specific analysis of GI disease parameters, such as serum tight junction (ZO-1, occludin, claudin) expression, serum LPS levels, and in vivo intestinal permeability assays to provide a connection between metabolic and microbial shifts and disease parameters (immune activation, hypersensitivity). Lastly, the authors did not describe why these sugars were chosen. IMO is a simple sugar, and FOS and GOS (like POL) have been shown by multiple studies to be associated with IBS and IBD symptoms, likely by increasing gas (H₂S, H₂) production following the global stimulation of anaerobic bacteria. I have left my major and minor comments below.

Major Comments:

- 1.) **Figure 1A:** Add markers to indicate when DSS treatment began, and when prebiotic treatment began. Additionally, you find a significant difference in body weight between DSS_ONLY and DSS_IMO, DSS_EEN, and DSS_FOS, but there was already a difference in body weight to begin (D7), so how can we be sure this difference at the end of the study is due to prebiotic treatment? Additionally, it appears prebiotic treatment did not elevate the body weight of these mice above EEN alone, which authors claim.
- 2.) **Figure 1B:** I only see a difference in DAI between FOS and EEN at D21, but again, there was already a difference between prebiotic treatment (D14). Thus, I am not confident prebiotic treatment and EEN improves DAI above EEN alone.
- 3.) **Figure 1C-1D:** Please arrange groups in bar plot (1D) in the same order as colon images (1C). EEN clearly has a longer colon than IMO and POL, but this is not reflected in the quantification. Please use representative images.
- 4.) **Figure 1E-1F:** Add control quantification here. I am surprised there is a significant difference between DSS and POL given their standard deviations overlap, ensure the correct statistical test was used. The POL treatment seems to have an irregular arrangement of cells, and the crypts are not easily discernable. Overall, there was no significant difference in histology score between EEN and EEN + prebiotics, as others claim.

5.) **Figure 2:** The alpha diversity between EEN and EEN + prebiotics is not significant. Beta-diversity differences seem to be minimal. I would want a comparison between EEN and prebiotic treatments, as that is the stated focus of the paper. Overall, this figure does not demonstrate there is a change in microbiome diversity between EEN and EEN treatment supplemented with carbohydrate prebiotics.

Minor Comments:

- 1.) Colors for various groups in Fig. 2 do not match rest of manuscript. Keep consistent.
- 2.) Statistical tests were used for Fig. 2A, 2B, and 2D but not 2C. Why? There seems to be a significant decrease in alpha diversity between CON and POL in Fig. 2B, but no significance is indicated. Is this correct?
- 3.) Fig. 4B: Can you add the groupings on top of the heat map for easier comparison?
- 4.) Page 2, line 41: "IMPORTANC"
- 5.) Pages 4-5, lines 87-91: Need reference
- 6.) Page 5, line 95: It would be helpful to some readers for a definition of EEN.
- 7.) Page 5, line 102: "Bifidobacterium" italicize all genus and species names throughout manuscript.
- 8.) Page 5, line 108: "probiotics" Please ensure you are using the correct term throughout manuscript.
- 9.) EEN consisted of propafenone. Is this a nutritive supplement? This is an antiarrhythmic medication.

- 10.) Page 9, line 182-184: What is "modeling"? I thought by the end of the second week (D14), all mice besides no-DSS controls were given 4% DSS, but authors refer to other groupings by this time point which confused me at first. Consider revising.
- 11.) Page 9, line 192: "In addition, he [the] colonic length"
- 12.) Page 10, line 207: "We compared the differences of [the] top 10..."
- 13.) Page 10, line 220: "*Lachnospiraceae Bacteroides A*" should be "*Lachnospiraceae 220 bacterium A4*"
- 14.) Page 12, line 252: "various mechanisms, including provide [providing] a better breeding space..." Consider alternative choice of words as bacteria do not "breed."
- 15.) Please change names of groupings in figures, as they all have DSS (except control), and all the prebiotic groups also have EEN.

Point-by-point response to the comments of the reviewers

Reviewer #2 (Comments for the Author):

N/A

Reviewer #4 (Comments for the Author):

Major concerns :

1. The abstract could be made more quantitative by including specific numerical data to support the results. For example, it could mention percentages of reduction in the Disease Activity Index (DAI), differences in colon length, or variations in microbial diversity indices (such as the Simpson index or the F/B ratio). This would better illustrate the magnitude of the effects observed and make the conclusions more convincing for the reader.

Reply 1: We thank the reviewer for their insightful comments and constructive suggestions, which have helped us to significantly improve the clarity of our manuscript. We added specific numerical data in the abstract to support the results, including disease activity index, colon length, histological score and F/B ratio. The revised abstract would read as follows: "The results revealed that EEN combined with prebiotic treatments showed more benefits in alleviating DSS-induced colitis than EEN treatment, among which FOS exhibited superior efficacy, including reducing disease activity index (DAI) (8.67 ± 0.52 VS 9.67 ± 0.82) and preventing colon shortening (7.23 ± 0.33 VS 6.43 ± 0.28 cm). Compared with the DSS group (7.00 [IQR: 5.75-8.25]), the histological score of IMO (2.50 [IQR: 2.00-4.00]), FOS (2.50 [IQR: 0.00-3.00]) and POL (2.00 [IQR: 1.50-7.25]) groups were obviously lower. Specifically, compared with the DSS group, IMO adjuvant treatment improved the F/B ratio (0.28 [IQR: 0.12-0.64] VS 1.06 [IQR: 0.87-2.82])."

2. Materials and methods : Detailed information on the origin, type, and source of the prebiotics used in the study is missing. Although the introduction mentions the names of the prebiotics (polydextrose, fructooligosaccharides and isomaltooligosaccharides) and their beneficial effects, it does not provide details on the exact type : Although the names of the prebiotics are given, there is no detailed description of their specific characteristics, such as their chemical structure or degree of polymerization, which could influence their effectiveness and the source or supplier: The study does not mention whether these prebiotics were purchased from a specific supplier or prepared in a laboratory, which is important information for the reproducibility of the results.

Reply 2: Thank you for your constructive comment on the details of prebiotics. The prebiotics in this study were all purchased from specific suppliers. In the chapter of "Materials and methods", we added the name of the supplier and the polymerization degree of prebiotics, as follows: "IMO (degree of polymerization is 3-7, manufacturer: Shanghai yuanye Bio-Technology Co., Ltd.), FOS (degree of polymerization is 3-7, manufacturer: Shanghai yuanye Bio-Technology Co., Ltd.) or POL (degree of polymerization is 20, manufacturer: Shanghai yuanye Bio-Technology Co., Ltd.)".

Specific comments:

3. Lines 24-25 : The destruction of gut microbiota is closely related to the occurrence and development of inflammatory bowel disease (IBD), which can be improved by the supplementation of prebiotics.

Comments : The term "destruction" may not be the most accurate in this context. It would be more appropriate to replace it with "disturbance" or "dysbiosis," as these terms better describe the imbalance or alteration in the gut microbiota composition associated with inflammatory bowel disease (IBD). The revised sentence would read:

"The disturbance (dysbiosis) of gut microbiota is closely related to the occurrence and development of inflammatory bowel disease (IBD), which can be improved by the supplementation of prebiotics."

Reference

Dysbiosis (disruption of the normal gut microbiota) is a hallmark of IBD, characterized by reduced bacterial diversity, decreased beneficial bacteria, and increased potentially harmful bacteria.

Hu, Y.; Chen, Z.; Xu, C.; Kan, S.; Chen, D. Disturbances of the Gut Microbiota and Microbiota-Derived Metabolites in Inflammatory Bowel Disease. *Nutrients* 2022, 14, 5140. <https://doi.org/10.3390/nu14235140>

Reply 3: We are grateful to the reviewer for the valuable feedback, which has helped us to significantly improve the manuscript. We have replaced the "destruction" with "disturbance" in the full text.

4. Line 38 : This study proved that probiotics as adjuvant therapy improved ... Please replace probiotics by prebiotics

Reply 4: Thank you very much for your professional comments. We were sorry for the mistakes in writing probiotics. We have replaced the "probiotics" with "prebiotics" in the full text.

5. Line 41 : Importance

Reply 5: We thank the reviewer for their thoughtful suggestions. We have corrected the word "Importance" in the manuscript.

6. Line 42 -44: Both sentences have some redundancy, as they both describe the benefits of prebiotics as an adjuvant treatment, albeit from slightly different angles. The first sentence focuses on clinical effects (body weight, colon length, ICD, histological score), while the second sentence focuses on microbiological effects (structure and composition of the gut microbiota).

Suggestion :

"We demonstrated the superior efficacy of prebiotics as adjuvant therapy in improving clinical outcomes (body weight, colonic length, DAI, and histological score) and modulating the structure

and composition of gut microbiota in DSS-induced colitis in mice.

Reply 6: We thank the reviewer for their constructive comments on our manuscript. We have incorporated all of them into the revised chapter of "Importance", and the specific contents were as follows: "We demonstrated the superior efficacy of prebiotics as adjuvant therapy in improving clinical outcomes (body weight, colonic length, DAI, and histological score) and modulating the structure and composition of gut microbiota in DSS-induced colitis in mice."

Minor comments :

7. The names of microorganisms are not always presented in italics, as is required by scientific conventions.

According to scientific standards, the names of genera and species of microorganisms should always be written in italics to respect taxonomic conventions.

Line 285 : *Prevotella*, line 286 : *Roseburia* Line 306 : *Lachnospiraceae* *Bacillus*, *Prevotella* SP Ma2016, and *Butyricimonas* SP Marcella; line 308 : *Butyricimonas*

Reply 7: We are grateful to the reviewer for the valuable feedback, which has helped us to significantly improve the manuscript. We have corrected the names of genera and species of microorganisms to italics in the full text.

8. References:

The quality of the references is as important as the quality of the manuscript itself and indicates the special care taken by the authors to submit a manuscript free of errors.

Reference section filled with errors (failure to italicize the scientific names of bacteria; article titles should be written in sentence style, without the capital letters throughout).

Reply 8: Thank you very much for your professional comments. We carefully checked and revised the references, including article titles and scientific names of bacteria.

Re: Spectrum01502-25R1 (Prebiotics as adjunctive treatment ameliorates DSS-induced colitis and gut microbiota)

Dear Dr. Xiaolin Ye:

Your manuscript has been accepted, and I am forwarding it to the ASM production staff for publication. Your paper will first be checked to make sure all elements meet the technical requirements. ASM staff will contact you if anything needs to be revised before copyediting and production can begin. Otherwise, you will be notified when your proofs are ready to be viewed.

Sincerely,
Xue Zhang
Editor
Microbiology Spectrum

Reviewer #4 (Comments for the Author):

Overall, the revisions have addressed all major and minor concerns raised by the reviewer.